# A Systematic Review on Password Guessing Tasks

**DOI:** 10.3390/e25091303

**Published:** 2023-09-07

**Authors:** Wei Yu, Qingsong Yin, Hao Yin, Wei Xiao, Tao Chang, Liangliang He, Lulin Ni, Qingbing Ji

**Affiliations:** 1The 30th Research Institute of China Electronics Technology Group Corporation, Chuangye Road, Chengdu 610093, China; 2Unit 95835 of PLA, Urumqi 841700, China; 3Wuhan Maritime Communication Research Institute, Canglong Avenue, Wuhan 430205, China; 4College of Computer, National University of Defense Technology, Deya Road, Changsha 410003, China; 5Northwest Institute of Mechanical and Electrical Engineering, Biyuan East Road, Xianyang 712000, China

**Keywords:** password guessing, trawling password guessing, targeted password guessing, deep learning

## Abstract

Recently, many password guessing algorithms have been proposed, seriously threatening cyber security. In this paper, we systematically review over thirty methods for password guessing published between 2016 and 2023. First, we introduce a taxonomy for classifying the existing methods into trawling guessing and targeted guessing. Second, we present an extensive benchmark dataset that can assist researchers and practitioners in successive works. Third, we conduct a bibliometric analysis to present trends in this field and cross-citation between reviewed papers. Further, we discuss the open challenges of password guessing in terms of diverse application scenarios, guessing efficiency, and the combination of traditional and deep learning methods. Finally, this review presents future research directions to guide successive research and development of password guessing.

## 1. Introduction

Information and communication technology represented by the Internet has profoundly changed people’s daily life. Although the application of the Internet provides convenience, cyber security risks inevitably follow. From a global perspective, the dangers posed by cyber security are becoming more and more prominent. They continue to penetrate various fields, such as politics, the economy, culture, society, and national defense. To protect the hardware, software, and data of the network system from being damaged, changed, or leaked, various cyber security protection methods are proposed. Among them, identity authentication plays a significant role in securing network systems.

The existing identity authentications can be broadly divided into the following four categories [1]:Based on what the user knows, such as passwords, PINs, etc.;Based on what the user owns, such as U-shields, encryption cards, etc.;Based on what the user is, such as human physiological characteristics (fingerprints, iris, face, voice, etc.) and user behavior characteristics (handwriting, mouse movement speed, keyboard tapping frequency, etc.);Multi-factor authentication technology, that is, multiple combinations of the above three authentications.

Figure 1 illustrates password-based authentication. The user is authenticated by entering a password through the public network to obtain the appropriate access rights. This kind of authentication is one of the most important methods to protect user information in the industry due to its strong practicability and ease of deployment [2].

With the continuous improvement of Internet application ecology, people have to set many passwords that satisfy different password policies to protect the corresponding software applications. As a result, each person has to remember many passwords in daily life. However, human memory capacity is relatively limited. Only five to seven passwords can be remembered [2]. Under this contradiction, users inevitably take measures to help memorize passwords, such as using weak passwords with low information entropy [3], using personal-related information in passwords [4,5], or reusing the same password in multiple systems [6,7]. Although these methods reduce the difficulty of password memorization, they are associated with serious security threats, allowing password guessing attacks to take advantage. Furthermore, a great deal of well-known websites’ password files have been leaked, which provides a constant source of material for password guessing to improve the attacking algorithm. In addition, with the prosperity of artificial intelligence, a large number of neural approaches for password guessing are emerging and the guessing ability has been continuously enhanced. In such circumstances, password security faces unprecedented pressure. Therefore, it is incredibly urgent and vital to sort out the research related to password guessing systematically. This review can present a reference for developers to design password authentication and provide password security researchers with a comprehensive understanding of the research dynamics in related fields. Table 1 lists all the abbreviations we make use of throughout this review.

### 1.1. Related Surveys and Reviews

Password guessing has multiple perspectives involving heuristic search, probabilistic models, and deep learning (DL). Some recently published survey papers have contributed overviews on password guessing, as Table 2 shows. In the following part of this section, we will present the related works in tabular order and compare them with our review at the end.

A password is a particular text sequence that has the characteristics of short length and rich semantics. Hence, taking full advantage of passwords’ semantics and language patterns is beneficial for password guessing. A comprehensive evaluation of semantic password grammar in terms of sample size, probabilistic smoothing, and linguistic information is presented in the survey [8], and it is compared with state-of-the-art Probabilistic Context-Free Grammar (PCFG) and neural network models in a cross-validation context. The experimental results reveal the contribution of syntax and semantic patterns to password guessing and show that the former has more impact on password security. Moreover, the paper illustrates that PCFG is often still competitive with the latest neural network counterpart models. A slight performance improvement has been achieved when training PCFG with more than 1 million passwords.

In the paper [9], the authors survey state-of-the-art deep learning methods for password guessing and password strength evaluation, including password pattern extraction, candidate password generation, and password strength measurement. However, due to its publication in 2020, the paper is only a statistics-relevant paper up to 2019. Various AI techniques have been rapidly iterated in recent years, and the paradigm has changed from models to machine learning theories. Therefore, there is an urgent need to update the relevant content.

A brief review of various existing typical password guessing algorithms in terms of assumptions, identification information, and theoretical models is presented in [10]. Multiple metrics are also used to understand and evaluate the performance of these algorithms. The characteristics of different password guessing algorithms are summarized by analyzing experimental results. It is demonstrated that two algorithms guess more passwords than one when the number of guesses is the same. In addition, the authors propose a hybrid password guessing algorithm, PaMLGuess, which has both strong interpretability and generalization ability and uses probability mapping to solve the problem that the probability sizes given by different password guessing algorithms vary greatly.

Artificial intelligence technology has advanced by leaps and bounds in recent years, blossoming and landing in many fields. Artificial intelligence is a double-edged sword. While it provides a variety of services to humans, threat actors use AI to conduct cyber attacks on society. Artificial intelligence is combined with traditional attack techniques to cause more damage. The paper [11] aims to explore existing research on AI-based cyber attacks and map it to a proposed framework that provides insight into new threats. The framework includes classifying several aspects of the malicious use of AI in the cyber attack lifecycle and provides a basis for its detection to predict future threats. Eleven case studies are found and classified into five categories: next-generation malware, voice synthesis, password-based attacks, social bots, and adversarial training.

A comparative analysis of deep learning algorithms and traditional probabilistic models for password guessing is presented in [12]. The authors analyze the password patterns of the leaked dataset and further present a comparative study of two mainstream probabilistic models, i.e., Markov-based and PCFG-based models, and the PassGAN model, which is a representative approach based on deep learning.

An empirical study of password management with 154 participants is reported in the paper [13]. The participants’ computers are instrumented to record detailed information about password characteristics and usage and many other computational behaviors. The study of these data enables a more accurate analysis of the password characteristics and behavior of the participants’ full web-based accounts. The findings include that the use of symbols and numbers in passwords predicts an increased likelihood of reuse, while increasing password strength predicts a decreased likelihood of reuse; password reuse is more common than previously thought, especially when partial reuse is considered; and password managers may have no effect on password reuse or strength. It is also observed that users could be grouped into a small number of behavioral clusters representing various password management policies. The findings suggest that, once users need to manage more passwords, they respond by partially and fully reusing passwords in most accounts.

A study that combined self-reported survey responses with measures of actual online behavior collected over six weeks from 134 participants is conducted in [13]. It is found that people tend to reuse passwords on 1.7–3.4 different websites, and most of them tend to reuse more complex and frequently entered passwords. The authors also investigate whether self-reported measures are accurate indicators of actual behavior and find that, although people understand password security, their self-reported intentions are only weakly correlated with reality. These findings suggest that users respond to the challenge of having many passwords by choosing a complex password to remember on a site.

The above reviews have different focuses on user habits, language patterns, AI threats, etc. Although some work attempts to compare deep learning with traditional probabilistic models, the selected deep models are homogeneous (i.e., GAN and RNN), and the statistical literature is very old. In this review, we collate the latest papers up to March 2023 and compare a large number of password guessing algorithms with different structures. Therefore, our work covers a broad range of methods, providing a holistic view of password guessing for password-based authentication.

### 1.2. Objectives and Contributions

A password is a widely used authentication method that aims to confirm the user’s identity through text matching. Although password-based authentication has some security and usability flaws and many new authentication techniques have been proposed, it will remain the dominant authentication method in the foreseeable future due to its simplicity, low cost, and ease of change. This work aims at providing an overview of recent neural approaches for password guessing, shaping the landscape of challenges and solutions in this field. Such a review is needed to help successive scientists and practitioners in the industry have a starting point for the research in password guessing.

The major contribution of this work are summarized as follows.

First, we classify password-guessing-related research into two categories according to the application scenarios: trawling password guessing and targeted password guessing. We compare and analyze those methods for researchers to understand the current status of password guessing.Second, we provide an extensive list of benchmark datasets for password guessing tasks so that interested researchers can easily find out baselines for their works.Third, we conduct an extensive bibliometric analysis from multiple perspectives to comprehensively depict the relationships among the existing password guessing studies.Finally, we discuss the open challenges of password guessing in terms of diverse application scenarios, guessing efficiency, and the combination of traditional and deep learning methods. Thus, this review helps to ease the efforts to find these requirements.

### 1.3. Paper Structure

The remainder of this paper is organized as follows. Section 2 outlines the methods for searching and selecting works for this review. Section 3 provides a summary of datasets for password guessing. Section 4 introduces a taxonomy to categorize password guessing methods and review the guessing method in detail. Section 5 presents the bibliometric analysis. Section 6 discusses the open challenges for this field. Section 7 provides the concluding remarks.

## 2. Research Method

This review for password guessing follows a systematic literature review methodology. We follow the procedure proposed by [14] to retrieve research papers from the existing literature, select relevant works out of the results, and summarize them afterward. Therefore, the systematic review process is reproducible and mitigates selection biases toward the works in the literature. Section 2.1, Section 2.2 and Section 2.3 outline research questions, the search strategy, and the study selection for the making of this review.

### 2.1. Research Questions

Research questions are the cornerstone of a literature review since every step of the review process relies on them. To come up with this review, we answer the following research questions:What are the current neural approaches for password guessing?How can the literature on password guessing be organized into a taxonomy that categorizes similar approaches based on the DL model, type of data, and application scenario?Which benchmark datasets are available for password guessing?What bibliometric relationships exist between the existing research studies?What are the open challenges regarding password guessing?

### 2.2. Search Strategy

The works we review in this paper were retrieved from top venues for NLP, machine learning, security, and privacy, which are indexed by either ACM Digital Library, IEEE Xplore, SpringerLink, Science Direct, or Web of Science. In addition to papers indexed by the aforementioned electronic scientific libraries, we also included valuable works in the e-Prints archive since this repository stores the most up-to-date research results. Once the list of electronic libraries was defined, we extracted search strings from the primary research questions and technical descriptions to retrieve relevant papers.

Table 3 lists the search terms we derived from the research questions and technical descriptions in two columns. The first seven rows of Term 1 are different descriptions of the password guessing task and the last two terms are the different guessing strategies. Term 2 contains technical terms. All 18 search terms are listed in the table combined to create search strings for the electronic libraries.

### 2.3. Study Selection

We applied the search terms in Table 3 to retrieve a total of 917 relevant works. Figure 2 shows a statistical histogram of the search results in each electronic library. Searches on ACM DL and IEEE Xplore returned most of the results, which account for 728 papers and 79% of the total. There is a certain amount of duplication and irrelevance in the retrieved works. Therefore, we must apply some inclusion and exclusion criteria to select the most relevant jobs. Published papers that satisfied all the following inclusion criteria were selected for this review.

Works that employed deep learning or traditional probabilistic method to model password guessability, such as PCFG, GAN, and VAE.Works published from the year 2016 onward. Since previous works have been reviewed in some works and this review aims to describe recent developments in the field of password guessing, we limited the publication time.Long papers or regular papers published in the main track of the conference or journal.Works published by top-tier venues, especially cybersecurity-related conferences or journals, such as S&P, CCS, NDSS, USENIX Conference, IEEE TDSC, or IEEE TIFS.

Published works that satisfied any of the following exclusion criteria were removed from this review.

Works published before 2016.Published works that did not contain the content of modeling password guessing.The experimental platform is not a desktop computer.Duplicated works.

After applying the above inclusion and exclusion criteria, we screened 41 papers that span from 2016 to March 2023.

## 3. Datasets

In this section, we provide an extensive list of benchmark datasets for password guessing tasks so that interested researchers can find out the basic information of baselines for their works. The list covers mainstream leaked password datasets and the datasets used in the reviewed papers. Since passwords are sensitive and private personal data, we cannot provide the relevant data or links directly.

As shown in Table 4, we summarized the dataset in seven aspects. The first column is the name of the leaked source system. A description of the services provided by the leaked source system is in the second column. The affiliation column and language column are statistics on which country the leaked source system is affiliated with and what official language is used. The fifth and sixth columns present the time and cause of the password leakage. The size column is used to describe the scale of the leaked dataset. The last column counts the times it was used in the reviewed papers.

Some preliminary conclusions can be drawn from the analysis of Table 4. First of all, most of the leaked data originated from well-known companies in large countries, such as the United States and China. Hence, English and Chinese are the dominant languages in the leaked password dataset. Second, most of the password data were leaked around 2010. Finally, in general, the larger the size of the dataset, the higher the probability of it being used as a research corpus. For example, the Rockyou is the largest English dataset in the table in terms of data size and it is also the most frequently used corpus in our reviewed works.

## 4. Password Guessing

Currently, the most serious security threat to password-based authentication systems is password guessing attacks. The research on password guessing has a long history and various methods and techniques have emerged. Based on different criteria, these studies can be classified into the following different categories.

According to whether the attack process requires interaction with the server, the password guessing attack can be classified into offline password guessing and online password guessing. The former attack requires the authentication server to store the user account password file and then the attacker makes a password guess on the local host. In this situation, the number of guesses that can be attempted is limited only by the attacker’s computing resources. The latter does not require a password file, and the attacker only needs to be connected to the network. However, the number of guesses that can be attempted is often limited by the server’s security policy, such as the US National Identity Standards NIST-800-63-3, which stipulates that the maximum number of failed logins allowed for a government website system in a month is 100, and the account will be locked if it exceeds 100 [15].According to whether the attack process utilizes the user’s personal information, password guessing attacks can be classified into trawling password guessing and targeted password guessing. While trawling guessing mainly exploits the user’s tendency to choose popular passwords, targeted guessing not only exploits the vulnerability of ordinary users to use popular passwords but also exploits the vulnerability of users to reuse passwords and construct passwords using personal information. With the help of information such as personally identifiable information and historical passwords, the success rate of targeted password guessing is significantly higher than trawling guessing with the same number of guesses.According to whether deep neural network methods of artificial intelligence are used in the attack process, password guessing attacks can be classified into neural attacks and traditional attacks. The former treats password guessing as a text generation task and relies on AI-related techniques to generate text passwords on a large-scale password training corpus. The latter encapsulates other methods, excluding deep neural network guessing methods.

In the remainder of this section, we divide the retrieved related work into trawling password guessing and targeted password guessing according to the second classification criterion above. In each category, we perform a secondary classification according to the third classification criterion to better compare and present the related works. The reason why the first taxonomy is not used is that almost all existing work is offline guessing and online guessing is too difficult to guess a password within 100 guesses.

### 4.1. Trawling Password Guessing

Trawling attacking aims to guess as many passwords as possible within the allowed guesses times and does not care who the specific target of the attack is. This means that the optimal attacker sequentially performs password guessing based on the ranking of the guessing probability. In this section, we review the existing methods in terms of both traditional trawling password guessing and neural trawling password guessing.

#### 4.1.1. Traditional Trawling Password Guessing Methods

Currently, heuristic search, Probabilistic Context-Free Grammar (PCFG) [16], and Markov sequence decision [17] are the mainstream backbone techniques for traditional trawling password guessing. In this section, the three methods mentioned above are introduced and analyzed.

**Heuristic Algorithms.** Early password guessing algorithms are basically trawling attack algorithms. These algorithms do not have a rigorous theoretical system and rely heavily on fragmented whimsy, for example, constructing unique guess dictionaries [18,19,20] using well-designed guess sequences [21] based on open-source software. As [3] points out, these heuristics are difficult to reproduce and to compare fairly with each other. Therefore, here, we only present some commonly used heuristic password guessing tools.

**John the Ripper (JTR)**, https://github.com/openwall/john (accessed on 20 June 2023). Originally, JTR was a password guessing tool that focuses on cracking UNIX/ Linux system weak passwords. There are four modes in JTR—single crack mode, wordlist crack mode, incremental mode, and external mode. Simple crack mode is specifically for users who use the account as the password. For example, an account username is ’admin’; the corresponding password may be ’admin888’ or ’admin123’. The wordlist crack mode requires the user to specify a dictionary file. Then, JTR performs password cracking based on the dictionary file. The incremental mode automatically tries all possible combinations of characters to be used as passwords. This mode is powerful but has a high time and space overhead. The external mode allows the user to develop a cracking module in C and then hook up the cracking module to the JTR environment to crack the password.**HashCat**, https://github.com/hashcat/hashcat (accessed on 20 June 2023). HashCat is the world’s fastest and most advanced password recovery utility, supporting five unique attack modes for over 300 highly optimized hashing algorithms. Hashcat currently supports CPUs, GPUs, and other hardware accelerators on Linux, Windows, and MacOS, and has facilities to help enable distributed password cracking.**L0phtCrack**, https://l0phtcrack.gitlab.io/ (accessed on 20 June 2023). The L0phtCrack project was created in 1997 by a group of hackers. After development and maintenance, the password auditing tool L0phtCrack was officially announced as open source. As a dedicated tool, L0phtCrack can be used to evaluate the strength of passwords and to help people in need recover lost passwords through brute force, dictionary, rainbow attacks, and other technical means.

There is also some analysis and optimization work for the above attack tools. In [22], the authors introduced techniques to reason analytically and efficiently about transformation-based password cracking in software tools like John the Ripper and Hashcat. They defined two new operations, rule inversion and guess counting, with which they analyze these tools without needing to enumerate guesses. Furthermore, they presented four applications showing how their techniques enable increased scientific rigor in optimizing these attacks’ configurations. In addition, work [23] discussed the measurement bias generated by using static policies for password strength estimation. To reduce this inherent measurement bias, the authors proposed a new generation of dictionary attacks called Adaptive Dynamic Mangling rules attack (AdaMs) that automates the advanced guessing strategies adopted by attackers and cast an adversary model that is consistently more resilient to inaccurate configurations to describe real-world attackers’ capabilities.

**PCFG.** In 2009, ref. [16] proposed the first fully automated trawling password guessing algorithm based on probabilistic context-independent grammar (PCFG). The core assumption of this algorithm is that the alphabetic segment *L*, numeric segment *D*, and special character segment *S* of the password are independent of each other. It first splits the password according to the three character types above. For example, “yu123456!” is split into the following structures—$L_{2}:yu,D_{6}:123456,and\;S_{1}:!$. The $L_{2}D_{6}S_{1}$ is called the pattern of that password. The algorithm includes training and guessing phases. In the training phase, the most critical thing is to calculate the frequency of password patterns (Structure) and character components (Semantic) based on the leaked password dataset. In the guessing phase, a set with frequency guesses is generated based on the pattern frequency table and semantic frequency table obtained in the training phase to simulate the probability distribution of the real password.

After years of development, some improved PCFG-based password guessing models have been proposed. Character is the primary processing unit of the traditional PCFG algorithm in password guessing. This processing granularity is too fine to obtain the semantic relationships between characters. To address this issue, ref. [24] extracted the semantic segments in the password based on word cohesion and freedom degree and improved the PCFG algorithm based on the semantic segments. Differently, in work [25], the authors regarded the password as a composition of several chunks, where a chunk is a sequence of related characters that appear together frequently to model passwords. This chunk segmentation method, called PwdSegment, extends the Byte-Pair Encoding (BPE) [26] algorithm by using the configurable parameter of an average length of chunk vocabularies to replace the number of the merge operations. In addition, to address the difficulty of long password guessing, ref. [27] proposed an improved PCFG-based long password guessing method named TransPCFG in which the knowledge learned by the short password PCFG algorithm is transferred to guess long passwords.

**Markov.** In 2005, ref. [28] introduced the Markov chain to password guessing for the first time. The core assumption of this algorithm is that users construct passwords sequentially from front to back. Instead of segmenting the password like PCFG, it trains the entire password and calculates the probability of the password by linking the characters from left to right.

The traditional Markov model is widely used in password guessing work due to its simple structure and fast inference. However, it also has certain drawbacks, such as overfitting, high repetition rate, and low coverage of the generated password based on random sampling. To address the above problems, some improved Markov-based password guessing models have been proposed.

To alleviate overfitting, ref. [17] applied Laplace smoothing and End-Symbol regularization techniques to the Markov model. The smoothing strategy eliminates the overfitting problem in the dataset, and the regularization technique makes the probabilities of the guesses generated by the attack algorithm always sum to 1.

To reduce the repetition rate of generated passwords, ref. [29] designed a dynamic distribution mechanism based on a random sampling method. This mechanism allows the probability distribution of the password to be dynamically adjusted and strictly converge to a uniform distribution during the guessing process. The authors proposed a dynamic Markov model based on the above dynamic distribution mechanism. Meanwhile, in order to model the semantic segments in the password, ref. [30] proposed a model named WordMarkov to extract the word cohesion and freedom degrees from the password through the semantic segments.

**Random Forest.** At USENIX’23, Wang et al. [31] proposed RFGuess, a random-forest-based framework that characterizes the three most representative password guessing scenarios. In the work, the authors assumed that the order in which users create passwords is from left to right, and each character is only related to a few characters before it. Under this assumption, the password generation process can be regarded as a multi-class classification problem. Assume that $T=\{\left(x_{1},y_{1}\right),\left(x_{2},y_{2}\right),\ldots ,\left(x_{n},y_{n}\right)\}$ is the training set, and then use the random forest to build a mapping *f* from the input space X to the output space Y. Here, X={n_orderstringsofapasswordset}, and Y={95printableASCIIcodes}∪{endsymbol}. The RFGuess is similar to the Markov 7-gram model [28]. Firstly, it processed the password into the form of 6-order character prefixes and their corresponding characters. Then, it represented the 6-order prefix as a 26-dimensional vector; the single character following this prefix in an ASCII value. During the training step, RFGuess traversed the 26-dimensional prefix feature vectors as training input and took the numerical label of the corresponding characters as training output.

#### 4.1.2. Neural Trawling Password Guessing Methods

With the rapid development of artificial intelligence, many studies have treated the password guessing task as a text generation problem and introduced some neural models in natural language processing.

In their pioneering work, in 2006, ref. [32] introduced shallow neural networks for password guessing. After years of development, in 2016, ref. [33] proposed FLA, which uses recurrent neural networks to estimate password distribution. This model follows the sequential decision procedure of the Markovian model but relaxes the n-markovian assumption. FLA can enumerate password space by a tree traversal algorithm to produce new guessing results. Since then, various generative neural networks based on have been observed repeatedly for trawling password guessing. In the following part of this section, we present the neural trawling password guessing methods by the neural model.

**Recurrent Neural Network (RNN).** As shown in Figure 3a, RNN is a recursive neural network that takes sequence data as input, recursion in the direction of sequence evolution, and all nodes (recurrent units) are connected in a chain. RNNs can be used not only as basic text generators but also can be embedded into other network as generative units. There are many variants of RNNs, such as LSTM [34], GRU [35], Bi-LSTM [36], etc.

The basic premise of the password guessing method using RNNs is that password guessing can be considered as a sequential text generation task. As shown in Figure 3b, the activations in the forward pass when the RNN is fed the characters *yu1* as input. The output layer contains confidences the RNN assigns for the next character (vocabulary is ‘*y,u1,2,3*’). We can see that, in the first time step, when the RNN saw the character *y*, it assigned a confidence of 1.0 to the next letter being *y*, 2.2 to the letter *u*, −3.0 to *1*, 4.1 to *2*, and 2.0 to *3*. Since in our training data (the password string *yu123*) the next correct character is *u*, we would like to increase its confidence (red) and decrease the confidence of all other letters (black).

The RNN-based password guessing methods usually have the following two steps:The password sequence from the training set is input to the RNN model for training in sequential text generation.The trained RNN guessing model aims to generate the next password character based on the existing password characters until the output termination character position.

Throughout the password generation process, the RNN calculates the probability of any character as the next password token. For a given threshold, a password with the probability that is higher than the threshold will be put into the password guessing set as a valid password.

Some researchers took RNN as a password generator with slight differences in the specific structure. Specifically, ref. [37] used the original RNN for password generation, and [38] segmented the password words and used Bi-LSTM to generate the password on the basis of word segments. In [39], the authors proposed a hierarchical semantic model called HSM, which combined LSTM with semantic analysis to mine the potential probabilistic relationships between words for password guessing. For Chinese passwords, ref. [40] proposed a password guessing method based on Chinese syllables. The method treated Chinese syllables as integral elements to parse and process passwords. And then, the processed passwords were trained in an LSTM neural network to generate the password.

Differently, in work [41], the authors used deep learning, specifically RNN with attention mechanism, to combine and interpolate information about users in the same group to define a robust and accurate prior over their password distribution. They presented the first self-configurable password model that uses auxiliary data to adapt to the target password distribution at inference time. In this way, a fully automatic approach to exploit auxiliary information and instantiate context-aware password meters and guessing attacks without requiring any plaintext samples from the target password distribution was developed.

**Generative Adversarial Network (GAN).** GAN [42] was proposed by Google researcher-Ian Goodfellow in 2014. It is an unsupervised learning method that unsupervised learning that makes the samples generated by the generative network obey the natural data distribution through adversarial training. As shown in Figure 4, the GAN-based password guessing model consists of a password generator and discriminator. The password generator takes random samples from the latent space as input, and its output needs to mimic the real password samples in the training set as much as possible. The input of the password discriminator network is the real password sample or the output of the generator network. The purpose of the discriminator is to distinguish the password output of the generator network from the real password sample. The two networks confront each other and continuously adjust the parameters. With the ultimate goal of making the discriminator network unable to determine whether the output of the generator network is real or not, so as to achieve the effect of password generation.

Further, ref. [43] proposed the first GAN-based trawling password guessing method called PassGAN in 2019, which does not rely on manual password analysis and autonomously learns the real password distribution from actually leaked passwords. Although the GAN-based guessing method provides a new way for trawling password guessing, there are several problems: the non-differentiability of discrete password data may lead to gradient back-propagation failure, the training of GAN-based password guessing model is difficult to converge, and the password generated by GAN model has a high repetition rate.

To address the problem of non-differentiability of the password discrete data sampling process, works [44,45] used Gumbel-Softmax [46] relaxation technique to train the GAN-based password guessing model. In addition, an alternative solution is provided in the work [44], which uses a smooth representation of the real password obtained by an additional autoencoder. In contrast, ref. [47] proposed RLPassGAN, a SeqGAN-based [48] password guessing method that uses policy gradients to ensure that the model parameters can be continuously optimized.

To alleviate the problem of convergence difficulties, ref. [49] designed a guessing algorithm based on a bidirectional generative adversarial network to improve the convergence rate. It can generate the same number of samples in a shorter time compared to the traditional GAN.

To deal with the problem of high repetition rate, ref. [47] argued that the root problem is that the output of the intermediate layer of the generator is an incomplete sequence of passwords that cannot be directly evaluated by the discriminator before reaching the output layer resulting in many redundant synthetic passwords. To address this problem, the authors proposed an improved method that uses Monte Carlo search [50] to evaluate the incomplete password sequences at the output of the intermediate layer. Differently, ref. [49] build an additional controller network using a discriminator and a controller to learn the metric between the generated password distribution and the true password distribution and uniform distribution, respectively. And then used these two metrics to teach the generator, thus reducing the repetition rate of password generation.

In addition to the above three issues, RLPass [51] innovatively used representation learning for password guessing. Specifically, the password is projected into the hidden space and the distance between password representations in the hidden space is used to define the similarity of the password. Based on the phenomena of strong locality and weak locality of passwords, the authors proposed a dynamic password guessing and conditional password algorithm. To address the low quality of long passwords generated by GAN-based models, ref. [52] designed a DenseNet-based [53] GAN structure for password guessing called DenseGAN, and two novel password guessing DenseGAN models are proposed, which both can generate high-quality password guesses.

**AutoEncoder (AE).** AE is an unsupervised learning model. It is based on back-propagation algorithms and optimization methods that use the input data themselves as supervision to guide the neural network to learn a mapping relationship to obtain a reconstructed output. The AE contains both encoder and decoder parts. According to the learning paradigm, AE can be classified as Undercomplete Autoencoder, Regularized Autoencoder, and Variational AutoEncoder (VAE), where the first two are discriminative models and the latter is a generative model. In the study of trawling password guessing, the Variational AutoEncoder [54] is usually used to generate password guesses.

As shown in Figure 5, the password sample (*yu123*) is input to the encoder of VAE to obtain the representation, and then the sample is reconstructed based on this representation using a decoder. The password generator is trained based on the reconstruction loss between the input *x* and generated $x^{\prime }$.

Although several trawling password guessing algorithms [47,51,55,56] use the VAE model to generate password guesses, each approach has a slightly different focus. In [55], the authors use the classical VAE framework to guess passwords without any change. Differently, ref. [47] combines VAE with the GAN technique and replaces the generator of GAN with VAE, aiming to solve back-propagation of discrete password data. Focusing on the light weight of the model, in [56], the complex RNN generation units are replaced by gated convolutional neural networks (GCNN) [57] to greatly reduce the complexity of the model.

**Transformer.** The Transformer model was proposed by Google in 2017 in the paper-*Attention Is All You Need* [58]. As shown in Figure 6, the Transformer model took the encoder–decoder architecture and used Attention [59] to replace the recurrent structure in the Seq2Seq model to parallelize sequence modeling. This parallelized structure provides a great shock to the field of natural language processing (NLP). As research progressed, related techniques gradually flowed from NLP to other fields, such as computer vision (CV), speech, biology, chemistry, etc. Similarly, there are some Transformer-based approaches in the research related to trawling password guessing [60,61].

The work [60] aims to study the probability of password cracking in common password rules and provide a reference for password setting. The authors collected a large number of users’ personal information and passwords and analyzed the correlation between personal information and passwords. Then, a password guessing model based on the improved Transformer was implemented. The work introduces message weights to data preprocessing and uses an improved beam search algorithm in the model to quickly search for the top-ranked password guesses. Differently, in [61], the authors proposed a bi-directional-Transformer-based guessing framework called PassBERT, which applied the paradigm of pre-training/fine-tuning to the password cracking for the first time. Specifically, first, the authors designed generalized password pre-training models that contain the knowledge of general password distributions. Then, three attack-specific fine-tuning approaches were proposed to tailor the pre-trained password model to the following real-world attack scenarios: conditional password guessing, targeted password guessing, and adaptive-rule-based password guessing. Finally, they further proposed a hybrid password strength meter to mitigate the risks from the three attacks.

**Reinforcement Learning (RL).** RL is one of the paradigms and methodologies of machine learning that is used to describe and address the problem of learning strategies to maximize reward or achieve specific goals by an intelligent agent during its interaction with the environment. Reinforcement learning theory, inspired by behaviorist psychology, focuses on online learning and tries to maintain a balance between exploration–utilization. Unlike supervised and unsupervised learning, reinforcement learning does not require any data to be given in advance, but, rather, learning information is obtained and model parameters are updated by receiving rewards (feedback) for actions from the environment.

As shown in Figure 7, the password generator is the agent, each generated password sequence represents a complete trajectory, and each character from the generation process is considered as an action. For example, a character generated by timestamp *t* is action $a_{t}$. Each action $a_{t}$ is generated according to a random policy on the current state $S_{t}$, which is determined by the generated incomplete sequence. The generator generates the password from arbitrary characters under a set of random policies until a predetermined length is satisfied. The discriminator is viewed as a reward function that evaluates the generated password sequence to update the parameters of the neural network.

In [47], the authors proposed a trawling password guessing model called RLPassGAN based on reinforcement learning and GAN. Specifically, the work follows the SeqGAN [48] to treat the password guess as a sequential decision and uses the policy gradient to ensure the parameters can be continuously optimized. Furthermore, the incomplete password sequences of the output are evaluated by Monte Carlo search [50]. Monte Carlo is a computational method that use a large number of random samples to learn about a specific system. It is very powerful and flexible, yet simple to understand and implement. It was proposed in 1940 in the *Manhattan Project*. The name came from the gambling city of Monte Carlo, symbolizing probability. In addition to the above applications, Monte Carlo has also been applied to estimate the guessing number of a given password [62,63].

**Flow.** Generative models GAN and VAE have been introduced in the above; neither of them explicitly learns the probability density function of real data p(x). In the case of generative models with latent variables, it is almost impossible to compute them p(x)=∫p(x|z)p(z) because it is difficult to traverse all possible values of the hidden variable *z*.

This challenge is solved by the generative model-normalizing flows [64], which is a powerful statistical tool for density estimation. One very distinctive feature of the flow model is that its transitions are usually reversible. As shown in Figure 8, the flow model not only finds a network pathway to transfer from distribution 
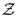
 to distribution 𝒳 but that pathway also allows 𝒳 to change to 
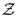
. In short, the flow model finds a duplex pathway between distributions 
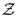
 and 𝒳. Of course, such reversibility comes at the cost that the data dimensions of 
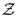
 and 𝒳 must be the same.

The transition between 
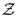
 and 𝒳 distributions is a tough job. The flow model goes through three steps. First, the Non-linear Independent Components Estimation (NICE) [65] achieves a reversible solution from the 𝒳 distribution to the Gaussian distribution. Later, RealNVP [66] proposes a reversible solution from the 𝒳 distribution to the conditional non-Gaussian distribution. Then, GLOW [67] designs a reversible solution from distribution 
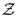
 to the 𝒳 distribution, where the 𝒳 distribution can be the same complex as the 
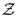
 distribution.

With the development of Flow, the Flow generative model has also been used for password guessing. In [68], Giulio Pagnotta et al. proposed PassFlow, a password guessing method based on a generative model of Flow. The Flow-based password guessing model uses exact log-likelihood computation and optimization to make the inference of latent variables accurate. In addition, a meaningful representation of the latent space is provided, which makes operations such as exploring specific subspaces of the latent space and interpolation possible. The authors demonstrate the applicability of the generative Flow model for password guessing. The experimental results show that PassFlow is able to outperform current GAN-based approaches in password guessing using a training set that is several orders of magnitude smaller than the previous method.

The above analysis reveals a wealth of research on trawling password guessing. Both traditional guessing methods and deep neural guessing methods have a large number of research. Since the datasets used in each work are different, it is difficult to make a cross-sectional comparison. Here, we take the widely used Rockyou dataset as an example to roughly illustrate the guessing effect of the existing methods. On the Rockyou dataset, the guessing accurate rate of the current trawling password guessing model is around 70% when the size of the guess set is $10^{9}$, and around 90% when the size of the guess set is $10^{12}$.

### 4.2. Targeted Password Guessing

For the given personal identifiable information (PII), the goal of the targeted password guessing attack is to guess the password as fast as possible [5,69]. Therefore, the attacker uses personal information related to the target person to enhance the guess. There are various kinds of personal identifiable information, such as demographic-related information (name, birthday, age, education, gender, etc.), expired passwords, and leaked passwords from other websites or systems. In this section, we follow the classification criteria in the above section to discuss targeted password guessing from traditional and neural models.

#### 4.2.1. Traditional Targeted Password Guessing Methods

**Targeted-Markov.** In 2015, ref. [69] first proposed a targeted attack guessing method based on a trawling Markov attack model [70]. The basic idea is that, regarding the percentage of the population that uses certain personal information, the target of the attack will also have the same likelihood percentage of using that personal information. To implement this idea, the literature [71] first divides the PIIs into several types, such as username-A, email prefix-E, and name-N, and further subdivides each broad type according to the desired granularity. Then, all PIIs in each password of the training set are replaced with the corresponding PII type. The remaining steps of the training phase are the same as those of the walking Markov model [71]. The guess set generation phase is divided into two steps. In the first step, the walking Markov model [71] is run to generate an intermediate guess set, which contains both directly usable guesses, such as *123456*, and intermediate guesses with basic characters of PII type (e.g., *N1*, *N2123*). The second step replaces the basic PII-type characters in the intermediate guesses with the corresponding PII information.

**Personal-PCFG.** In 2016, ref. [5] proposed a targeted attack guessing method based on PCFG, called Personal-PCFG. It follows the trawling PCFG attack model [16]. The basic idea is the same as the PCFG attack model: slice the password by character type and length. To implement this idea, the literature [5] divides the PIIs into six major types (i.e., username-A, email prefix-E, name-N, birthday-B, phone number-P, and ID-G) and treats these six PI character types as equal to L, D, and S in the trawling PCFG model so that there are nine types of characters in the Personal-PCFG. Then, during the training process, each password in the training set is segmented by the corresponding character type and its length as in the trawling PCFG attack model [16].

**TarGuess** Wang et al. [4] proposed a framework that systematically characterizes typical targeted guessing scenarios with four mathematical probabilistic models. The first scenario, TarGuess-I, aims to utilize the PII of the user to create an online targeted password guessing. To represent the PII tokens in the password, the authors defined 28 PII labels based on the type of personal information (e.g., $N_{1}-N_{7}$ and $B_{1}-B_{10}$), in addition to the L, D, and S labels in the PCFG [16] model. For each PII label, its subscript number represents a subdivision of the PII usage of this type instead of the subscript number indicating the corresponding length, as in the case of L, D, and S labels. For example, N represents the name information, while $N_{1}$ represents the full name and $N_{2}$ represents the abbreviation of the full name. The second, Targuess-II, aims to guess the user password in a target website (e.g., CSDN) given the user’s leaked password in other websites (e.g., Dodonew). Specifically, the authors propose six structural-level and two character-level mnemonic transformations to depict password reuse and a Markov model is used to carve a context-independent transformation grammar based on the above reuse rules. The third, TarGuess-III, aims to guess the user’s password using a sister password and some PII information. TarGuess-III introduces PII information to the TarGuess-II model, allowing PII information to be embedded in structure-level password reuse. Compared to TarGuess-III, the attacker in the TarGuess-IV scenario knows additional PII that is difficult to quantify (e.g., gender, education). To solve the problem that some of the PII is difficult in terms of being directly embodied in the password, the authors cleverly introduce Bayesian theory to compute the reuse probability of the password based on this difficult-to-quantify PII.

**RFGuess-PII.** Based on RFGuess introduced in Section 4.1.1, Wang et al. [31] proposed a new targeted password guessing model called RFGuess-PII. The password training and generating process is similar to the trawling guessing scenario. The difference is that the PII string in the password is replaced with the corresponding digital tag through the new PII matching. This new PII matching aims to minimize the information entropy and tries to accurately extract the PII usage behavior of the entire user group. The first step of the PII matching algorithm is to subdivide the various possible transformations of PII and use digital tags to represent them. The second step is to list all the possible representations with PII tags for each password in the training set. Then, the representations were sorted by frequency from high to low.

**RFGuess-Reuse.** In addition to PII-based targeted password cracking research, in [31], the authors also focused on modeling users’ password reuse behavior. They also considered both structure-level and segment-level transformations, like TarGuess-II [4]. Specifically, they count structure-level transformations by calculating the editing matrix for each password pair in the training set, and a segment-level transformation (i.e., a transformation within a string of the same type, e.g., *password* → *passwor* in letter segment) model based on random forest was trained.

**TG-SPSR.** In [72], the authors transpose both Markov and PCFG models to targeted attacks and propose a systematically targeted attack model based on structure partitioning and string reorganization, called TG-SPSR. In the structure partition phase, in addition to dividing the password into a basic structure similar to PCFG [16], the authors define a trajectory-based keyboard in the basic syntax pattern and introduce index bits to accurately describe the location of special characters. In addition, a BiLSTM classifier is constructed to reuse and modify the behavior of passwords based on the defined nine modification rules.

#### 4.2.2. Neural Targeted Password Guessing Methods

With the development of natural language processing techniques, some sophisticated neural networks are applied to the field of targeted password guessing. In [73], the authors proposed a targeted password guessing model called PG-Pass composed of the pointer generator network. This work innovatively considers targeted password guessing as a summarization task and applies pointer network techniques commonly used in the field of intelligent summarization to it.

It should be noted that, in addition to the user’s demographic-related information (name, birthday, etc.), the user’s leaked passwords on other websites can also be exploited by attackers to perform targeted attacks. It can be expected that such targeted attacks that exploit the vulnerable behavior of user password reuse may be more harmful than attacks based on demographic-related information.

Based on the fact that users are always eager to generate new passwords by reusing or fine-tuning old ones, ref. [74] proposed a password Transformer-based reuse model and simulated credential tampering attacks. At the IEEE S&P’19 conference, Pal et al. [75] introduced deep learning techniques to characterize users’ password reuse behavior. More specifically, they trained a sequence-to-sequence (seq2seq) model to predict the modifications required to transform an existing password to their sister passwords and carried out validation on a large-scale dataset (i.e., 4iQ dataset [76]). In addition, Wang et al. [77] proposed a targeted password guessing algorithm, PASS2EDIT, to model the increasingly damaging credential tweaking attack, in which an attacker exploits the victim’s leaked passwords to increase her success rate of guessing the victim’s passwords at other sites. Particularly, they proposed a multi-step decision making training mechanism and built a classification neural network to learn the reaction of one-step edit operations to the existing password.

Since the datasets used in each work are different, it is difficult to make a cross-sectional comparison. Taking the 12306 dataset as an example, the existing targeted password guessing methods have a guess accuracy rate of about 41.07% when the guessing set is 100 [73]. In the reuse guessing scenario, when the victim’s password at site A (namely $pw_{A}$) is known, within 100 guesses, the cracking success rate of the SOTA method [77] in guessing her password at site B ($pw_{B}=pw_{A}$) is 24.2% (for common users) and 11.7% (for security-savvy users), respectively.

## 5. Bibliometric Analysis

Bibliometrics is a cross-science that analyzes all knowledge carriers quantitatively using mathematical and statistical methods. It is a comprehensive system that integrates mathematics, statistics and bibliography and focuses on quantification. To better present the quantitative relationships among the reviewed papers, we conduct a bibliometric analysis in this section.

### 5.1. Statistics of Technology

To present the development trend of password guessing research, we perform detailed technical statistics in this section. In Section 5.1, we present comprehensive statistics of the various techniques and models used in the reviewed papers.

#### Fine-Grained Technical Statistics

To present a comprehensive view of the categories, adopted technologies, and release times of the reviewed papers, we design a fine-grained technology statistics chart as shown in Figure 9. The names (the correspondence between method name and paper is detailed in Appendix A Table A1.) of password guessing methods in the reviewed papers in descending order by proposed time are shown on the left side of Figure 9. The timeline and the number of technologies are presented on the right and below Figure 9, respectively. In the following part, we will analyze and explain the technical statistics from a different dimension.

From the perspective of time, it can be seen that a small amount were published sporadically from 2016 to 2018 and the publication number in 2021 increased to nine. This quantitative increase indicates the overall research on password guessing shows a rising trend. In terms of the technology distributed location, the guessing methods based on deep neural networks are mainly concentrated in the upper left corner of the figure, while the guessing methods based on traditional methods are mainly concentrated in the lower right corner. This phenomenon indicates that the research on neural password guessing has been hot in recent years. It also reflects that the focus of password guessing technology is changing from the traditional method to the deep neural network method.

From the perspective of numbers, the techniques used in neural password guessing methods are more diversified in many studies. Among them, GAN, RNN, and CNN are the most-used techniques. For traditional password guessing methods, the PCFG and Markov are the most widely used guessing techniques.

From the perspective of categories, the existing research mainly focuses on trawling password guessing with as many as 37 guessing techniques, and there are only six works about targeted password guessing research. This indicates that the existing studies are more concerned with trawling password guessing, and even less research has been conducted on targeted password guessing.

Although many parties are very concerned about password-related research, password guessing research is still not perfect and mature. On the one hand, as mentioned earlier, there is a lack of systematic research on password guessing attacks, and most of them focus on individual attack scenarios, such as offline trawling guessing. further, the increasingly realistic targeted online guessing has rarely been addressed. On the other hand, even though a few studies focus on targeted password guessing, they mostly stay at the level of simple statistics on user behavior and reliance on heuristic whims, lacking theoretical and principled research. Some fundamental problems need to be solved.

### 5.2. Cross-Citation Analysis

In order to investigate the cross-citation relationships among the reviewed papers, we conducted a cross-citation analysis in this section. As shown in Figure 10, we used a chord chart to visualize the cross-citation relationships. The left part of the chord chart reflects the number of papers cited by others, and the right half indicates the number of papers citing others. The middle arc represents the existence of a cross-citation relationship between two papers. A closer look at the chord chart can easily reveal that FLA [33] and PassGan [43] are the most-cited works by others. This indirectly illustrates the influence of the two works. Since FLA and PassGAN are the first methods to use RNN [34] and GAN [42] to model password guessability, respectively, it can confirm why RNN and GAN are used in high frequency.

To further explore the cross-citation of the reviewed papers, we conducted a statistical analysis in terms of the nationality of the institution to which the first author belongs. As shown in Figure 11, we colored the country regions on the world map with the number of publications. At a glance, it is clear that the Chinese region has the deepest coloring. It means that China is the country that published the most papers in the password guessing field. The United States follows China in second place. Looking further at the graph, we can observe an interesting phenomenon that almost all citations revolve around these two countries with a large number of publications. The arrows in the figure represent the citation direction, and the numbers on the arrows represent the number of citations. For example, the number 35 on the arrow points to the United States from China. This means that twenty-five papers published by China cite six papers published by the United States a total of 46 times. To quantitatively estimate the influence of the country in the field of password guessing, we follow the impact factor (IF) method in the Journal Citation Reports produced by Thomson Reuters to calculate. Specifically, the impact factor of each country is calculated as follows: $IF=\frac{Citations}{Publications}$. After calculation, the IF of China is 0.28 and the IF of the United States is 11. This value indicates that, although China has a large number of publications, its influence in this field is far less than the United States, which has a smaller number of publications.

## 6. Challenges and Future Trends

Despite the progress made in password guessing, many issues remain to address. In this section, we point out the main challenges and future trends that emerged from the analysis presented in this paper.

### 6.1. Diverse Application Scenarios

The existing research mainly focuses on the scenario of trawling guessing, but, in real application scenarios, there are often the requirements of targeted guessing, few-shot guessing, and low resources. These application scenarios are real and highly demanded. Taking the few-shot targeted password guessing as an example, it means that the attacker can guess the password of a given user in a given service according to relevant targeted information (user personal information, system password policy, etc.). For this scenario, the existing methods have difficulty solving the problem of few-shot. Although techniques such as data augmentation can be used to alleviate the lack of samples, it is still tricky to fundamentally break through the dilemma. Likewise, various deep learning algorithms also rely heavily on the size and quality of the samples, and the collection of sensitive information such as passwords is difficult. Therefore, guessing targeted passwords against limited password samples is a complicated problem.

### 6.2. Improving Guessing Efficiency

The existing password guessing methods are designed without considering the space–time complexity. However, in real applications, time and computing resources are often limited. Both traditional and neural password guessing methods require enormous computing power to support. As the neural network models become increasingly complex, the storage requirements also increase tremendously. Therefore, quickly performing password guessing with low resources is also challenging. In the reviewed papers, only one work discussed a lightweight password guessing model [56]. Thus, a great deal of work in this area still needs to be advanced urgently.

### 6.3. Combing Traditional and Deep Learning Methods

As the analysis in Section 5 shows, the techniques used in password guessing are relatively outdated compared to related research. With the rapid development of technology, as researchers of password guessing, we should embrace emerging technologies to give full play to traditional methods. As a typical generative task, password guessing should theoretically pay close attention to the latest developments in the field of intelligent generation. However, looking at the existing research, emerging generative models, such as flow [64] and diffusion models [78,79,80], are rarely applied in password guessing. In addition, the current neural password guessing methods have a premise that the password string is a sequence of characters, and the sequence encoding is used to process the password. In fact, we can make more attempts, such as organizing passwords into graphs according to their semantic dependencies and using the latest graph neural network (GNN)-related technologies [81,82,83,84] for sampling and generation. Moreover, the pre-trained language models [85,86,87,88] have been in full swing in the field of natural language processing in recent years. However, only one work [61] has discussed the paradigm of pre-training/fine-tuning regarding password guessing. Therefore, it might also be a good idea to combine the powerful general language modeling capabilities of pre-trained language models with the existing password guessing efforts.

## 7. Conclusions

In this paper, we present a systematic review for modeling password guessability, covering datasets, methodology, and bibliometric analysis. We propose a taxonomy structure to classify the existing password guessing works as a guide to readers, with the advantage of enabling this structure to be extended to future works and neural methods. Our review covers over 37 methods for password guessing published between 2016 and 2023, which crack passwords from the perspective of guessing scenario: trawling and targeted.

To summarize, we investigated the question of how to guess the password. Firstly, we approached this endeavor by considering different scenarios and divided into traditional and neural categories for comparison and discussion. Secondly, we collected extensive benchmark datasets to assist researchers and practitioners in successive works. Finally, we conducted an extensive bibliometric analysis to present trends in this field and cross-citation between reviewed papers.

## Figures and Tables

**Figure 1 entropy-25-01303-f001:**
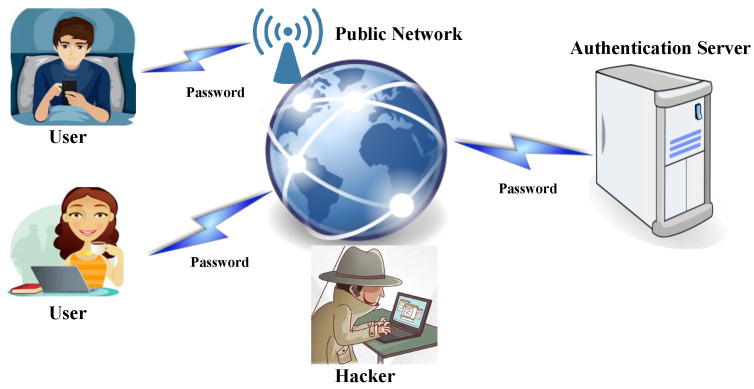
Schematic diagram of password-based authentication.

**Figure 2 entropy-25-01303-f002:**
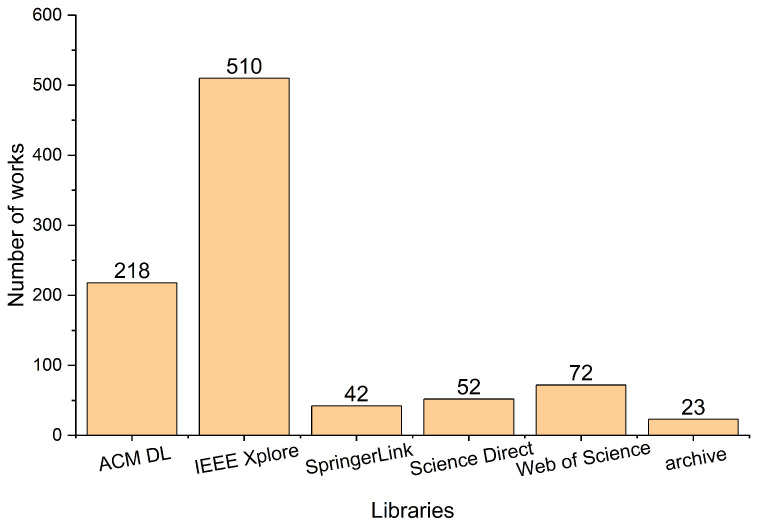
Search results from electronic libraries.

**Figure 3 entropy-25-01303-f003:**
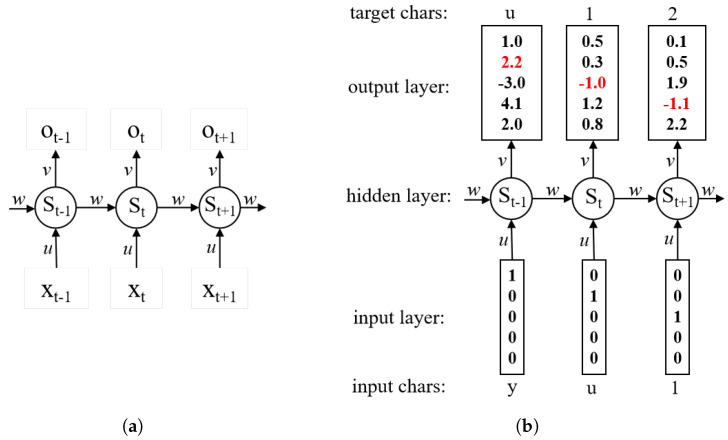
An example of basic RNN architecture and an RNN-based char-level password guessing model with 5-dimensional input and output layers. (**a**) The basic RNN architecture; (**b**) An illustration of RNN-based char-level password guessing.

**Figure 4 entropy-25-01303-f004:**
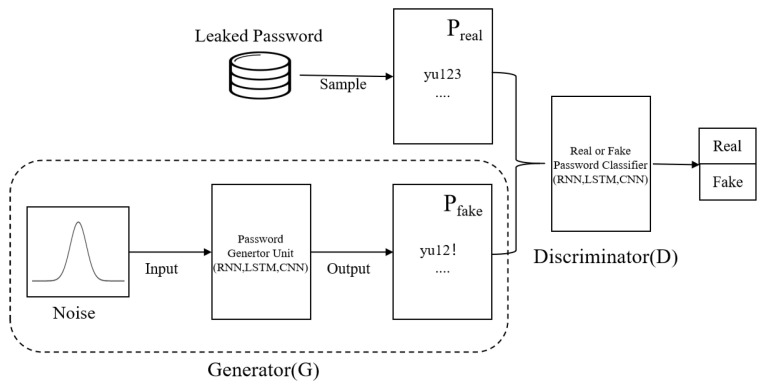
An illustration of GAN-based password guessing model.

**Figure 5 entropy-25-01303-f005:**
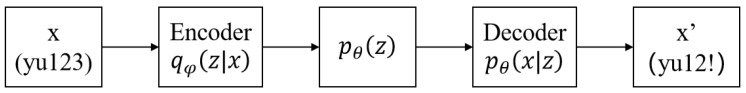
An illustration of VAE-based password guessing model.

**Figure 6 entropy-25-01303-f006:**
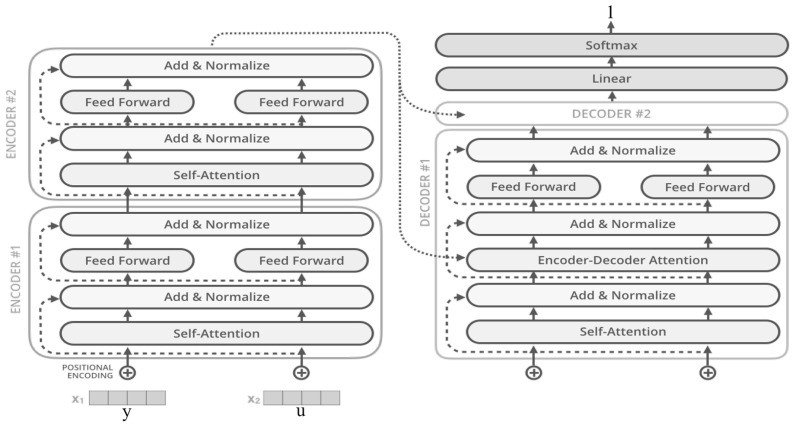
An illustration of Transformer-based password guessing model.

**Figure 7 entropy-25-01303-f007:**
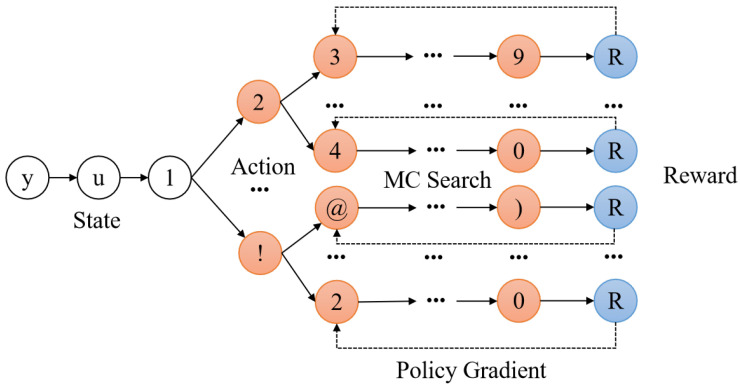
An illustration of reinforcement-learning-based password guessing model.

**Figure 8 entropy-25-01303-f008:**
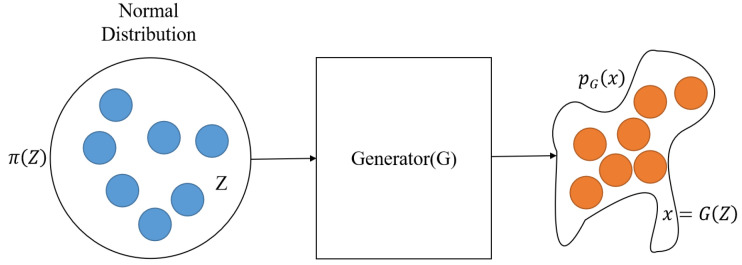
An illustration of flow-based password guessing model.

**Figure 9 entropy-25-01303-f009:**
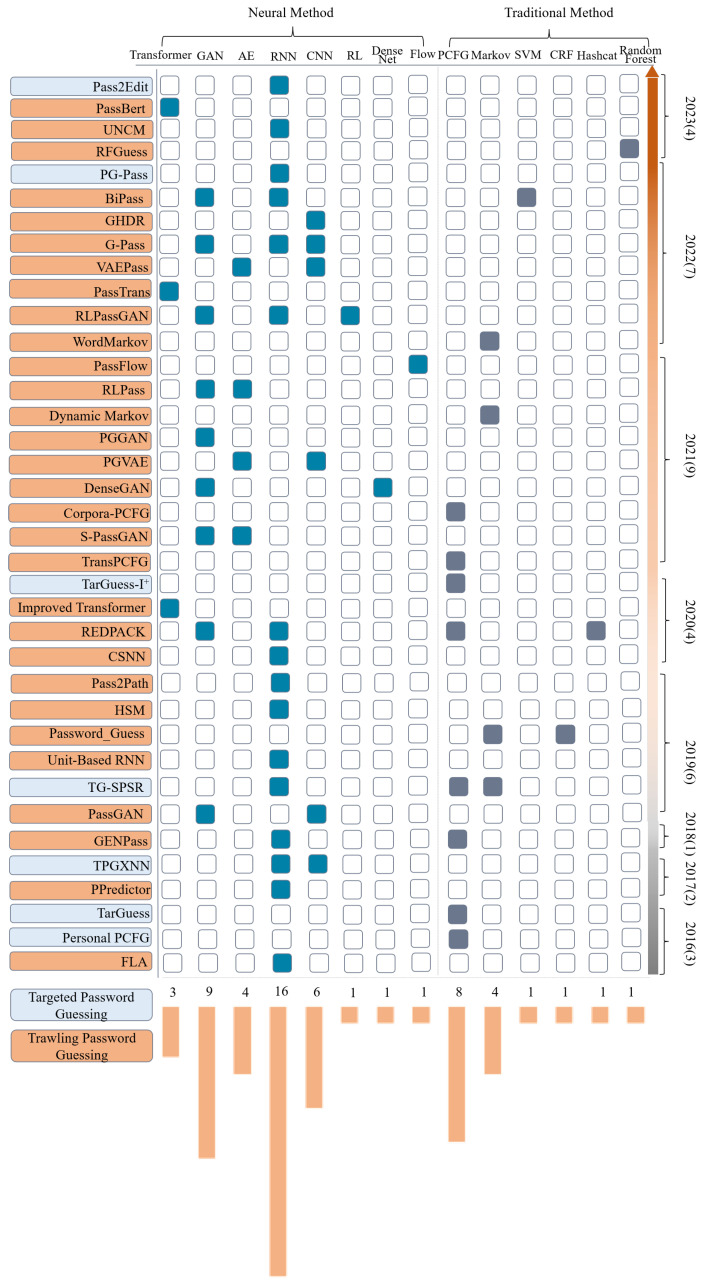
A schematic of the fine-grained technical statistics of the reviewed papers.

**Figure 10 entropy-25-01303-f010:**
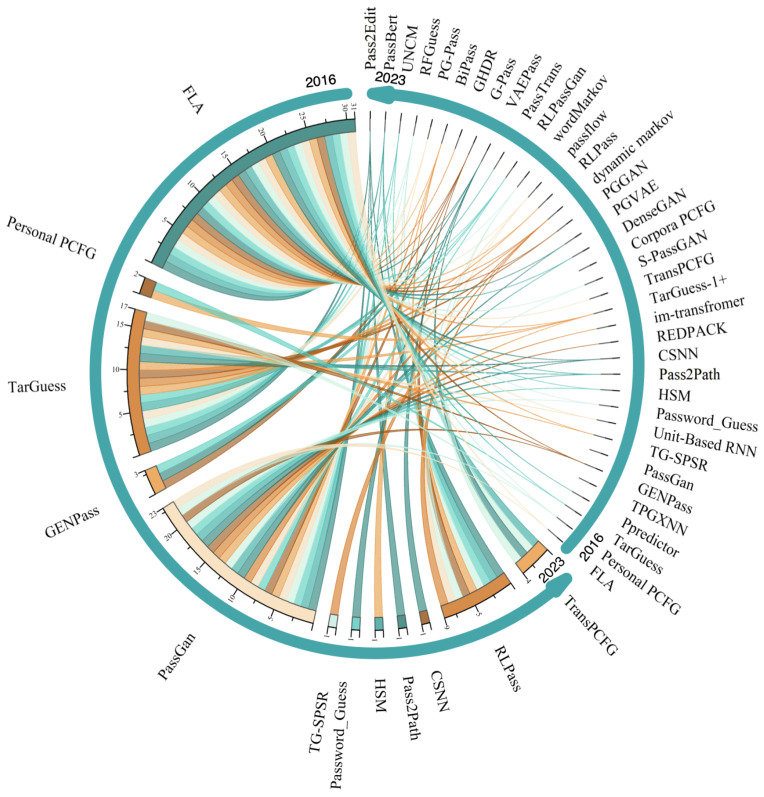
A schematic diagram of cross-citation relationships among reviewed papers.

**Figure 11 entropy-25-01303-f011:**
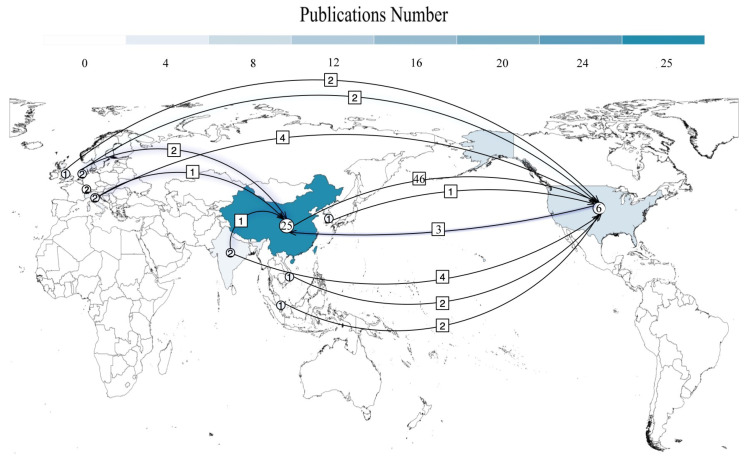
An illustration of cross-citation by country. Circled numbers represent the number of relevant papers published in that country, and boxed numbers indicates the number of citations by other countries.

**Table 1 entropy-25-01303-t001:** Abbreviations used throughout this review.

Abbreviation	Description
AE	Auto-Encoder
CNN	Convolutional Neural Network
CRF	Conditional Random Field
DL	Deep Learning
GAN	Generative Adversarial Network
NLP	Natural Language Processing
PCFG	Probabilistic Context-Free Grammar
RL	Reinforcement Learning
RNN	Recurrent Neural Network
SVM	Support Vector Machine
PII	Personal Identifiable Information

**Table 2 entropy-25-01303-t002:** Overview of related surveys on password guessing research.

Work	Year	Contributions
A Large-Scale Analysis of the Semantic Password Model and Linguistic Patterns in Passwords [8]	2021	Provided a comprehensive evaluation of semantic password grammars in terms of sample size, probabilistic smoothing, and linguistic information.
Deep Learning for Password Guessing and Password Strength Evaluation: A Survey [9]	2020	Surveyed state-of-the-art deep learning methods for password guessing and password strength evaluation before 2019.
A Preliminary Analysis of Password Guessing Algorithm [10]	2020	Took the Coverage metric to quantify the involvement of personal information in the creation of an individual password and used the Monte Carlo and zxcvbn methods to assess password strength.
The AI-Based Cyber Threat Landscape: A Survey [11]	2020	Explored research examples of cyber attacks posed by combining the “dark” side of AI with the attack techniques and introduced an analytic framework for modeling those attacks.
Deep Learning vs. Traditional Probabilistic Models: Case Study on Short Inputs for Password Guessing [12]	2019	Focused on the comparative analysis of deep learning algorithms and traditional probabilistic models on strings of short-length passwords.
Let’s Go in for a Closer Look: Observing Passwords in Their Natural Habitat [13]	2017	Shown that password reuse in both exact and partial form is extremely rampant.

**Table 3 entropy-25-01303-t003:** Password guessing terms used to create search expressions.

Term 1	Term 2
password guessing	neural network
password cracking	neural approach
password attacking	machine learning
password guessability	deep learning
password predictability	VAE
password reuse	Transformer
password probability model	PCFG
trawling	GAN
targeted	Markov

**Table 4 entropy-25-01303-t004:** Statistics of the leaked passwords datasets.

Name	Type	Affiliation	Language	Timestamp	Causes of Leakage	Size	Times
Tianya	Social Website	China	Chinese	Dec. 2011	Hacker Attacks	30,233,633	3
Dodonew	Games and E-commerce	China	Chinese	Dec. 2011	Hacker Attacks	16,231,271	8
CSDN	Programmer Forum	China	Chinese	Dec. 2011	Hacker Attacks	6,428,287	14
12306	Railway Website	China	Chinese	Dec. 2014	Crash Attack	131,653	6
126	Email	China	Chinese	Oct. 2015	Crash Attack	136,711,126	2
163	Email	China	Chinese	Oct. 2015	Crash Attack	118,100,272	2
17173	Games Website	China	Chinese	Dec. 2011	Crash Attack	18,333,810	0
Rockyou	Games	USA	English	Dec. 2009	SQL Injection	32,603,388	18
000webhost	Free Web Hosting	USA	English	Oct. 2015	Programming Errors	15,251,073	8
Battlefield	Games Website	USA	English	Jun. 2011	Hacker Attacks	417,453	0
Single.org	Dating Website	USA	English	Oct. 2010	SQL Injection	16,250	0
Faithwriters	Writing Forum	USA	English	Mar. 2009	SQL Injection	9709	0
Hak5	Hacking Forum	USA	English	Jul. 2009	Hacker Attacks	2987	0
Myspace	Social Website	USA	English	May 2016	Hacker Attacks	30,213,024	5
Twitter	Social Website	USA	English	May 2012	Hacker Attacks	22,805,966	2
Gmail	Email	Russia	Russian	Sep. 2014	Phishing Attacks	4,929,090	2
Mail.ru	Email	Russia	Russian	Sep. 2014	Phishing Attacks	4,932,688	0
Yandex.ru	Search Engine	Russia	Russian	Sep. 2014	Phishing Attacks	1,261,810	0
Flirtlife.de	Marriage Website	Germany	German	May 2006	Hacker Attacks	115,589	0

## Data Availability

No new data were created or analyzed in this study. Data sharing is not applicable to this article.

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
