# Peer review of "A Systematic Review on Password Guessing Tasks"

_entropy, 2023, doi:10.3390/e25091303_

Round 1
Reviewer 1 Report
This paper provides an overview of existing techniques used for estimating passwords, offering insights into the various tools and research advancements in this field. Additionally, it explores additional meta aspects of the paper, including citation chains and the geographical distribution of the authors.
The primary concern I have with the paper is its omission of several crucial works, despite considering a substantial number of research items. Although the authors claim to have conducted a "bibliometric analysis," some significant contributions are conspicuously absent. These are the ones I could spot:
- [S&P’2019a] Reasoning Analytically about Password-Cracking Software
- [S&P’2019b] Beyond Credential Stuffing: Password Similarity Models Using Neural Networks
- [USENIX’21] Reducing Bias in Modeling Real-world Password Strength via Deep Learning and Dynamic Dictionaries
[S&P’2019] and [USENIX’21] are particularly relevant as they cover dictionary attacks which are the most widely employed tool used in real-world attacks. The survey is lacking in that direction. Similarly, [S&P’2019b] is one of the main works on targeted attacks. It is very strange that the paper does not mention it.
The following are more recent works, but I think they would make the paper more comprehensive. Especially [S&P’24] that posses between Targeted and Untargeted attacks:
- [USENIX’23] Improving Real-world Password Guessing Attacks via Bi-directional Transformers
- [S&P’24] Universal Neural-Cracking-Machines: Self-Configurable Password Models from Auxiliary Data
I find section "5.1.2. Lag Analysis of Technology" to be misleading as it suggests that newer machine learning models are inherently superior for password security, while older ones are inherently inferior. However, this is not necessarily true. For example, excluding dynamic password guessing, PCFG methods, or expert dictionary attacks have demonstrated better performance than PassGAN. The performance of a password guessing method is not solely determined by its age; there is little correlation between the "dotage" of a method and its effectiveness. In fact, properly configured standard methods often outperform modern ones. I believe this part of the paper requires revision to accurately reflect the relationship between method age and password guessing performance.
The English in the paper sounds odd and there are a bunch of issues with document format. For instance, the abstract appears two times.
Some of the sentences do not make particular sense to me. For instance:
- “Published works that did not use intelligent method to model password guessing“ Do you mean machine learning methods?
- “These algorithms do not have a rigorous theoretical system and rely 297 heavily on fragmented whimsy. “
- “JTR is a password guessing tool that attempts to crack 302 plaintext when the ciphertext is known.”
Author Response
Point-by-Point Response to Reviewers
Manuscript ID: Entropy-2497948
Manuscript Title: A Systematic Review on Password Guessing Task
Dear Editor,
Thank you for your positive evaluation on our manuscript and allowing us a revision of our manuscript with an opportunity to address the reviewers' comments.
In the following, we provide a point-by-point response to the comments and the corresponding manuscript changes. As below, the reviewer's comments are written in black and our responses in blue. The essential amendments or modifications to the manuscript are given after the response in red.
Best regards,
Dr. Wei Yu
Dear Reviewers,
Many thanks for your positive evaluation on our manuscript and your careful consideration and valuable comments, which have helped us improve the manuscript significantly. We have fully addressed the concerns raised and incorporated the suggestions, hoping that we have addressed them satisfactorily in the revised manuscript, as indicated in the responses below.
Reviewer #1:
We are highly grateful for your time and professional review work on our manuscript and encouraging comments/suggestions. According to your nice suggestions, we have made the required modifications/corrections to our previous draft, the detailed corrections are listed below and also included in revised manuscript.
Comment 1:
The primary concern I have with the paper is its omission of several crucial works, despite considering a substantial number of research items. Although the authors claim to have conducted a "bibliometric analysis," some significant contributions are conspicuously absent. These are the ones I could spot:
- [S&P’2019a] Reasoning Analytically about Password-Cracking Software
- [S&P’2019b] Beyond Credential Stuffing: Password Similarity Models Using Neural Networks
- [USENIX’21] Reducing Bias in Modeling Real-world Password Strength via Deep Learning and Dynamic Dictionaries
[S&P’2019] and [USENIX’21] are particularly relevant as they cover dictionary attacks which are the most widely employed tool used in real-world attacks. The survey is lacking in that direction. Similarly, [S&P’2019b] is one of the main works on targeted attacks. It is very strange that the paper does not mention it.
The following are more recent works, but I think they would make the paper more comprehensive. Especially [S&P’24] that posses between Targeted and Untargeted attacks:
- [USENIX’23] Improving Real-world Password Guessing Attacks via Bi-directional Transformers
- [S&P’24] Universal Neural-Cracking-Machines: Self-Configurable Password Models from Auxiliary Data
Author response:
We are thankful to reviewers for highlighting the shortcoming of our manuscript. We are sorry for any confusion caused by the incomplete collection and retrieval of papers. We have revised the manuscript according to the reviewer’s suggestions. In the revised manuscript, we have added reviews and analyses of the important work listed by the reviewer. In addition, to make our study more comprehensive, we also re-searched the papers related to password cracking at the four major cybersecurity conferences(S&P, USENIX, CCS and NDSS) in the past five years, line by line.
Change in the manuscript: The following changes have been made in the revised manuscript. We have added 10 password cracking papers to the revised manuscript. Moreover, all charts in the bibliometric analysis section were updated according to the additions.
These are listed by the reviewer:
- [S&P’2019a] Reasoning Analytically about Password-Cracking Software
- [S&P’2019b] Beyond Credential Stuffing: Password Similarity Models Using Neural Networks
- [USENIX’21] Reducing Bias in Modeling Real-world Password Strength via Deep Learning and Dynamic Dictionaries
- [USENIX’23] Improving Real-world Password Guessing Attacks via Bi-directional Transformers
- [S&P’24] Universal Neural-Cracking-Machines: Self-Configurable Password Models from Auxiliary Data
These are the ones I could spot:
- [USENIX’23] Password Guessing Using Random Forest
- [USENIX’23] PASS2EDIT: A Multi-Step Generative Model for Guessing Edited Passwords
- [S&P’23] Confident Monte Carlo: Rigorous Analysis of Guessing Curves for Probabilistic Password Models
- [CCS’21] Chunk-Level Password Guessing: Towards Modeling Refined Password Composition Representations
- [CSCWD’22] PG-Pass: Targeted Online Password Guessing Model based on Pointer Generator Network
We have organized the above papers into the revised manuscript and updated the dataset and bibliometric statistics sections accordingly.
Comment 2:
I find section "5.1.2. Lag Analysis of Technology" to be misleading as it suggests that newer machine learning models are inherently superior for password security, while older ones are inherently inferior. However, this is not necessarily true. For example, excluding dynamic password guessing, PCFG methods, or expert dictionary attacks have demonstrated better performance than PassGAN. The performance of a password guessing method is not solely determined by its age; there is little correlation between the "dotage" of a method and its effectiveness. In fact, properly configured standard methods often outperform modern ones. I believe this part of the paper requires revision to accurately reflect the relationship between method age and password guessing performance.
Author response:
We are very thankful to the reviewer for his/her valuable comments to improve the quality of our manuscript. According to the reviewer’s suggestion, all authors engaged in a discussion and concluded that the situation described by the reviewer existed. However, our original intention was to show that the latest techniques have rarely been tried in password guessing and to encourage people to be brave enough to try new approaches. To eliminate this misleading, we have deleted the section.
Change in the manuscript: we have deleted the section 5.1.2 .
Comment 3:
The English in the paper sounds odd and there are a bunch of issues with document format. For instance, the abstract appears two times.
Some of the sentences do not make particular sense to me. For instance:
- “Published works that did not use intelligent method to model password guessing“ Do you mean machine learning methods?
- “These algorithms do not have a rigorous theoretical system and rely 297 heavily on fragmented whimsy. “
- “JTR is a password guessing tool that attempts to crack 302 plaintext when the ciphertext is known.”
Author response:
We are very thankful to the reviewer for his/her valuable comments to improve the quality of our manuscript. According to the reviewer’s suggestion, we further embellished the paper and adjusted the format of the article.

Reviewer 2 Report
The given Systematic Review on Password Guessing consider methods published between 2016 and 2023. A taxonomy for classifying the existing methods into trawling guessing and targeted guessing is introduced and an extensive benchmark dataset is discussed. A bibliometric analysis is performed and open challenges of password guessing are stated. The review certainly is interesting, updated and possily useful.
The proposed taxonomy is roughly the following:
* Trawling Password Guessing
- Traditional Trawling Password Guessing Methods
- Neural Trawling Password Guessing Methods
Recurrent Neural Network (RNN)
Generative Adversarial Network (GAN)
AutoEncoder (AE)
Transformer
Reinforcement Learning (RL)
* Targeted Password Guessing
- Traditional Targeted Password Guessing Methods
Targeted-Markov
Personal-PCFG
TG-SPSR
- Neural Targeted Password Guessing Methods
The taxonomy has the purpose to classify the existing password guessing works as a guide to readers. The review covers methods for cracking passwords with respect to guessing scenario: trawling and targeted.
It would be interesting as well to review the possible attack methods and their corresponding defense strategies. The system administrators are mos interested in protecting their systems than in biblioteconomical studies regarding password cracking.
English is good and readable.
Author Response
Point-by-Point Response to Reviewers
Manuscript ID: Entropy-2497948
Manuscript Title: A Systematic Review on Password Guessing Task
Dear Editor,
Thank you for your positive evaluation on our manuscript and allowing us a revision of our manuscript with an opportunity to address the reviewers' comments.
In the following, we provide a point-by-point response to the comments and the corresponding manuscript changes. As below, the reviewer's comments are written in black and our responses in blue. The essential amendments or modifications to the manuscript are given after the response in red.
Best regards,
Dr. Wei Yu
Dear Reviewers,
Many thanks for your positive evaluation on our manuscript and your careful consideration and valuable comments, which have helped us improve the manuscript significantly. We have fully addressed the concerns raised and incorporated the suggestions, hoping that we have addressed them satisfactorily in the revised manuscript, as indicated in the responses below.
Reviewer #2:
Comment 1:
It would be interesting as well to review the possible attack methods and their corresponding defense strategies. The system administrators are mos interested in protecting their systems than in biblioteconomical studies regarding password cracking.
Author response:
We are highly grateful for your time and professional review work on our manuscript and encouraging comments/suggestions. We have seriously considered the reviewer's suggestions. Still, since the defense of password attacks is a systematic research that covers many aspects (password strength assessment, authentication protocols, and so on), it is difficult to incorporate this piece into our review in the limited time available for revision. In our future work, we will conduct an extensive study on the defense against password attacks.

Reviewer 3 Report
This is an excellent survey and analysis of the recent papers on password guessing algorithms; in my opinion this paper could be viewed as an excellent tool for any researcher in this field, but especially as a starting point for anyone new to this field. I would strongly recommend it to PhD students, for example.
Current trends and open challenges are also covered. Very intuitive figures are included, that greatly improve the readability and the understanding of the material.
Author Response
Point-by-Point Response to Reviewers
Manuscript ID: Entropy-2497948
Manuscript Title: A Systematic Review on Password Guessing Task
Dear Editor,
Thank you for your positive evaluation on our manuscript and allowing us a revision of our manuscript with an opportunity to address the reviewers' comments.
In the following, we provide a point-by-point response to the comments and the corresponding manuscript changes. As below, the reviewer's comments are written in black and our responses in blue. The essential amendments or modifications to the manuscript are given after the response in red.
Best regards,
Dr. Wei Yu
Dear Reviewers,
Many thanks for your positive evaluation on our manuscript and your careful consideration and valuable comments, which have helped us improve the manuscript significantly. We have fully addressed the concerns raised and incorporated the suggestions, hoping that we have addressed them satisfactorily in the revised manuscript, as indicated in the responses below.
Reviewer #3:
Comment 1:
This is an excellent survey and analysis of the recent papers on password guessing algorithms; in my opinion this paper could be viewed as an excellent tool for any researcher in this field, but especially as a starting point for anyone new to this field. I would strongly recommend it to PhD students, for example.
Current trends and open challenges are also covered. Very intuitive figures are included, that greatly improve the readability and the understanding of the material.
Author response:
We are highly grateful for your time and professional review work on our manuscript and encouraging comments/suggestions.

Round 2
Reviewer 1 Report
I am satisfied by the revision.
I suggest to the authors to perform another round editorial round on the paper.